# Measurement Method of Temperature of the Face Gear Rim of a Spiroid Gear

**DOI:** 10.3390/s22228860

**Published:** 2022-11-16

**Authors:** Łukasz Chodoła, Aleksander Mazurkow, Mirosław Surowaniec, Tadeusz Markowski, Wojciech Homik

**Affiliations:** 1Faculty of Mechanical Engineering and Technology, Rzeszow University of Technology, Kwiatkowskiego 4, 37-450 Stalowa Wola, Poland; 2Department of Machine Design, Faculty of Mechanical Engineering and Aviation, Rzeszów University of Technology, Al. Powstańców Warszawy 8, 35-959 Rzeszów, Poland

**Keywords:** spiroid gear, face gear, worm, thermal imaging camera, gear casing, types of spiroid gears, higher torques, lubrication of spiroid gears

## Abstract

Spiroid gears are used to transfer heavy loads with a significant reduction in input speed. Like most toothed gears, they are lubricated with oil whose physical properties change with temperature fluctuations, affecting the durability and reliability of the gear. Bearing this in mind, gear designers plan systems for measuring oil temperature during gear operation at the design stage. The authors of this paper are of the opinion that, in the case of spiroid gears, it may be insufficient to measure only oil temperature during gear operation. It seems that the working temperature of a pair of mating wheels has a decisive impact on the durability and reliability of the gear. The measurement of oil temperature in a tested gear should be treated as a supplementary measurement with the measurement of temperature on the toothed wheels as the basic measurement. Taking into consideration the above, an innovative test bench was designed and built, making it possible to observe how working parameters of the gear (torque and rotational speed) affect the temperature of the lubricating oil, but most of all, the working temperature of the pair of mating wheels. This paper presents, among others, the results of research on the impact of the rotational speed of the input shaft and load on the distribution of temperature on the toothed rim of the face gear.

## 1. Introduction

Worm gears are used in case of operation of drive systems when the change in the rotational speed entails a significant change in the torque. Spiroid gears (Figure 1), which have been increasingly used in drive systems of various mechanisms, are an alternative to worm gears.

For example, they are used as follows:In the aviation industry as mechanisms for flight control and positioning of radar antennas;In the transport industry as drive systems for electric vehicles, elevators, escalators, and mechanisms supporting work of actuators;In the arms industry (Figure 2) as mechanisms for controlling gun position, opening and closing the hatch, doors, and devices replacing hydraulic systems.

Both worm gears and spiroid gears have their axes skewed. Advantages of spiroid gears over worm gears with the same gear ratio include the following:Higher efficiency and silent operation;Possibility to achieve a higher torque on the output shaft;More favorable bearing lubrication conditions;Higher number of face gear teeth in contact with the worm threads. This effect is achieved through the use of a spiral tooth line of the face gear; as a result, the pressure points are distributed over a larger contact area;Design of the spiroid gear allowing for the bearings to be placed close to the meshing; consequently, spiroid gears are stiffer compared to worm gears [1].

Worm gears have a line contact, whereas spiroid gears have a point contact. Both types of gears have a common disadvantage: they may heat up.

The results of operational and experimental research show that abrasive wear, deformation, and fatigue wear of meshes substantially contribute (approx. 30%) to the loss of gear durability. High oil temperatures accelerate the wear process [2].

In oil-lubricated low-power gears, there is no need to check oil quality. Oil change intervals are determined by the gear manufacturer [3]. In case of high-power and high-torque gears, oil quality checks are recommended. Oil change intervals depend on the physical and chemical properties analysis.

Since no research study dedicated to this type of gears was found in the available literature, a test bench was designed and built, enabling, among others, innovative measurement of temperature on the toothed rim of the face gear mating with the worm and measurement of the oil temperature. Temperature was measured with use of thermocouples and a thermal imaging camera. This paper presents the test results and their analysis. Further research is currently underway, and the results will be presented in subsequent publications.

## 2. Construction and Operation of a Gear

A spiroid gear (Figure 3) consists of two major components: a worm and a face gear. In this type of gear, the shaft is combined with the worm. In the case of speed reduction gears, the worm works as a driver. The worm winding is cut along the helix. In this type of gear, the worm axis is shifted from the face gear axis by a > 0.5 d_z2_, where d_z2_ is the outer diameter of the face gear. The applied solution helps to keep more teeth of the face gear in contact, which increases the contact area of the mating rims. In addition, it also enables smoother motion transmission and higher torques on the output shaft.

There are two types of spiroid gears: left-hand and right-hand gears. The hand of a gear is determined by the direction of the worm shaft threads in relation to the direction in which the face gear rotates. Figure 4 presents the types of spiroid gears depending on the direction of the worm shaft threads and the tooth line of the face gear.

As mentioned before, spiroid gears may be an alternative to be used in mechanisms that require a high gear ratio, high positioning accuracy, and self-locking, such as dividing heads, table drives of the hobbing machine, or motor reducers in belt feeders. Various designs of spiroid gears are described in [1,3,7,8,9,10,11,12,13,14,15]. The design and calculation methods applicable to the discussed gears are presented in [13,16,17,18,19,20,21,22,23,24,25], whereas their application in various drive systems are presented in [13,26].

## 3. Lubrication of the Mating Toothed Rims of a Gear

Spiroid gears are lubricated with oils [27]. During operation the surfaces of the mating teeth roll around and slide against each other. The friction force increases the temperature of the mating teeth and the oil. A choice of oil considerably affects the operation of a gear [2,27]. Table 1 presents the dependence between the peripheral speed of the worm and the lubrication method. 

For lubrication of spiroid gears, it is recommended to use oils with anti-seizure additives (EP—extreme pressure) [29,30]. Under the conditions of elastohydrodynamic friction, such oils facilitate the formation of an oil film between the contacting tooth surfaces. 

In practice, the following three spiroid gear lubrication methods are used for industrial purposes: manual lubrication, immersion lubrication, and spray lubrication. 

Manual lubrication is intermittent lubrication (Figure 5), which consists in applying the lubricant at specific intervals. Intermittent lubrication is usually used where the peripheral speed of the worm is up to 2 m/s [28]. A choice of the lubrication method may also be determined by the dimensions of the gear, its casing, and safety considerations. 

When the peripheral speed of the worm is higher than 2 m/s (Table 1), it is recommended to use immersion or spray lubrication (Figure 6 and Figure 7).

Immersion lubrication uses the rotation of the worm and face gear immersed in oil. The worm tooth should be immersed in oil to the depth of ≈3 m, where “m” is the normal module. The rotating input shaft delivers oil to the mating worm threads and the worm wheel teeth. The splash delivers oil to the rolling bearings. Where the peripheral speed of the worm exceeds 8 m/s, the spray lubrication method should be used (Figure 6) [29].

## 4. Methods for Measuring Temperatures in a Gear

The requirements for gears, such as work with higher loads and speeds, increase the requirements for the mating toothed rims. This implies a need for new design methods that better reflect the real operating conditions. These methods are based on bench tests supported by theoretical models.

The analyzed literature provides no descriptions of test benches or results of measurements of the parameters describing the operation of a gear. In order to ensure correct operation of a gear, one must know how temperature is distributed in the meshing areas. The measurement of temperature in the oil tank is insufficient because it is determined as the average temperature resulting from the heat balance of the heat generated in the friction pairs and the heat discharged to the environment.

Therefore, an original measurement method was proposed, taking into account the type of gear and dimensions of the toothed rim of the face gear. The subject method uses the following:Thermocouples located on the face gear rim;Thermal imaging camera installed inside the body of the gear.

## 5. Test Bench Used to Measure the Temperature in the Meshing of the Toothed Wheels

Figure 8 presents an innovative test bench for measuring the mechanical and thermal properties of spiroid gears. It enables the measurement of:Torque on the input shaft (Nm);Torque on the output shaft (Nm);Rotational speed on the input shaft (rpm);Oil temperature at the gear inlet (°C);Oil temperature in the oil tank (°C);Temperatures in six test points located on the face gear rim;Temperatures in the meshing area with the use of a thermal imaging camera.

Temperature in the face gear rim is measured with the use of thermocouples. The thermocouples were mounted in six holes evenly spaced around the circumference made on the front side of the face gear. The location of thermocouples is presented in Figure 9 and Figure 10.

The electric wires of the thermocouples were led outside the gear casing through a hole made on the shaft on which the face gear is mounted. The entire measuring unit is closed by joint located on the front side of the shaft.

The temperature field of the face gear was also measured with the use of a thermal imaging camera mounted inside the gear casing. The measurements were performed using Optris Xi80, a miniature thermal imaging camera with focus adjustment and electric drive. The basic dimensions of the camera are presented in Figure 11.

The measurement results were analyzed with use of dedicated software. Figure 12 presents the location of the camera in the casing. The advantage of this type of camera is that it operates autonomously and collects information on temperatures in a specific area in real time [31]. The accuracy of the camera measurement according to the manufacturer is ±2 °C.

## 6. Bench Test of Spiroid Gear

The bench tests of the temperature distribution in the measuring area of the meshing of the worm and the face gear were carried out for spiroid gear whose selected mechanical and geometric parameters are presented in Table 2.

Table 3 presents the measurement results for individual measurement areas, gear loading torque, and rotational speed of the input shaft.

Temperature distribution in the front rim for example measurements Table 3 is included in Figure 13, Figure 14 and Figure 15.

## 7. Measurement Results

The tests were carried out for two speeds of the input shaft, 500 and 1500 rpm, and four resistance torques, M_0_ = 0, 35, 90, and 350 Nm.

The distributions of temperature in the face gear rim presented in Figure 13, Figure 14 and Figure 15 point to the following conclusions:An increase in the shaft rotational speed and the resistance torque causes an increase in the meshing temperatures;The change in the temperature along the tooth line of the face gear is insignificant;The temperatures in the tooth space are lower than in the addendum;Face gear tooth spaces are not evenly heated;The temperature of the face gear toothed rim increases along with an increase in rotational speed and load.

## 8. Summary

This paper presents an innovative test bench for testing spiroid gears. This is innovative research and has not been reported so far, as found from the available literature. The test bench enables temperature measurements at test points located on the face gear near the tops of the teeth, as well as in the oil tank and the body of an operating gear. In addition, the thermal imaging camera installed in the body of the gear allows for measuring temperature distributions on the toothed rim of the face gear. The bench tests were carried out and showed how changes in the rotational speed of the input shaft and resistance torque affect the distribution of temperature in the toothed rim of the face gear. The tests were carried out for a gear with a gear ratio u = 20.

Based on an analysis of the test results, it was found that changes in the rotational speed of the input shaft of the gear and changes in the resistance torque directly impact the temperature of the oil in which the gear operates and the map of temperatures in the toothed rim of the face gear (Table 3). The examples of test results presented in Table 3 prove that both the temperature of the oil and the temperature of the toothed rim of the face gear change with the increase of resistance torque.

For the value of resistance torque changing from 0 to 35 Nm in the range of tested speeds, the increase in temperature was approximately 14 °C and approximately 5 °C on the toothed rim of the face gear. It is noteworthy that, in extreme cases, the difference between the temperature of the oil and the temperature of the toothed rim of the face gear did not exceed 8 °C.

The results of the bench tests where the gear was loaded with higher torques (Table 3, measurements 5, 6, 7, and 8) showed a significant increase in the temperature of both the oil and the toothed rim of the face gear. In the tested load range of the gear, the change in the temperature of the oil and the toothed rim of the face gear was approximately 18 °C and 26 °C, respectively. The largest difference between the temperature of the oil and the temperature of the toothed rim of the face gear amounted to approximately 16 °C and was observed at the highest gear load (Table 3, measurement 8).

It is a known fact that significant values of gear ratios and torques increase the values of Hertz stresses and resistance to motion; in such cases, gears may operate with mixed or fluid friction (thermo-elasto-hydrodynamic friction). It happens when oil and its properties play a vital role. By lubricating the teeth of the toothed rims, the oil reduces the resistance to movement, while by flowing through the meshing zone, it lowers the temperature of the teeth and removes contamination.

The bench tests showed that the evaluation of the temperature of a working gear based only on the temperature in the oil tank may be insufficient, because an excessive increase in the temperature of the face gear or worm teeth may cause thermal wear and tear to the gear and reduce its durability.

The next stage of the research will involve developing a thermo-elasto-hydrodynamic model of the oil film in the meshing of toothed wheels and conducting durability tests of the gear. These tests will enable the determination of verified static characteristics of the oil and the development of a verified algorithm for the design of this type of gears.

## Figures and Tables

**Figure 1 sensors-22-08860-f001:**
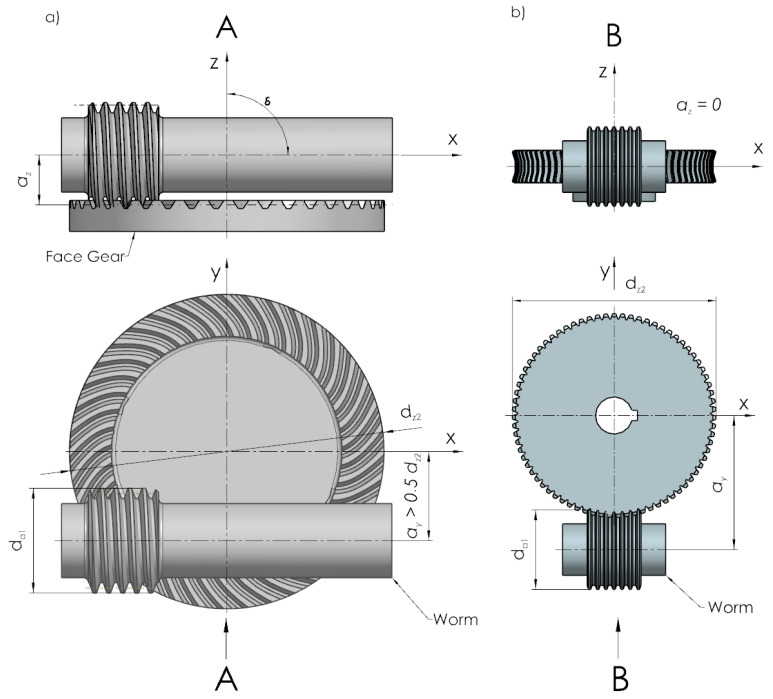
Location of wheels in (**a**) spiroid gear and (**b**) worm gear.

**Figure 2 sensors-22-08860-f002:**
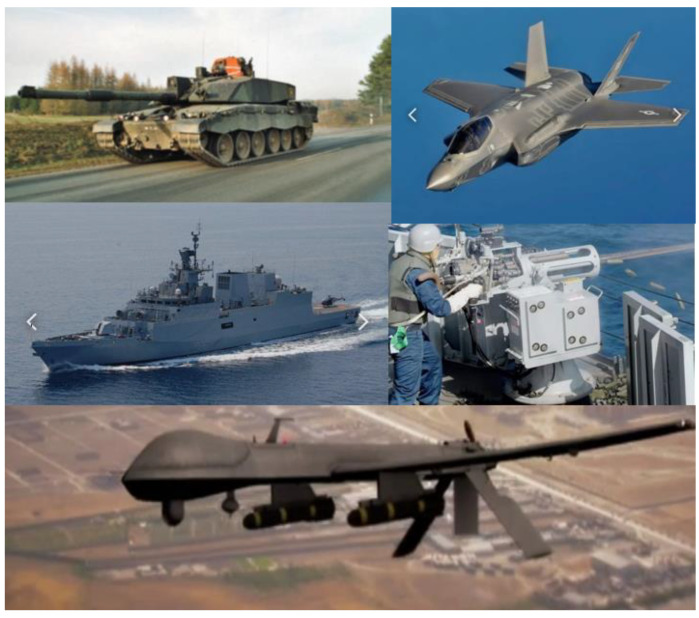
Exemplary applications of spiroid gears [4,5,6].

**Figure 3 sensors-22-08860-f003:**
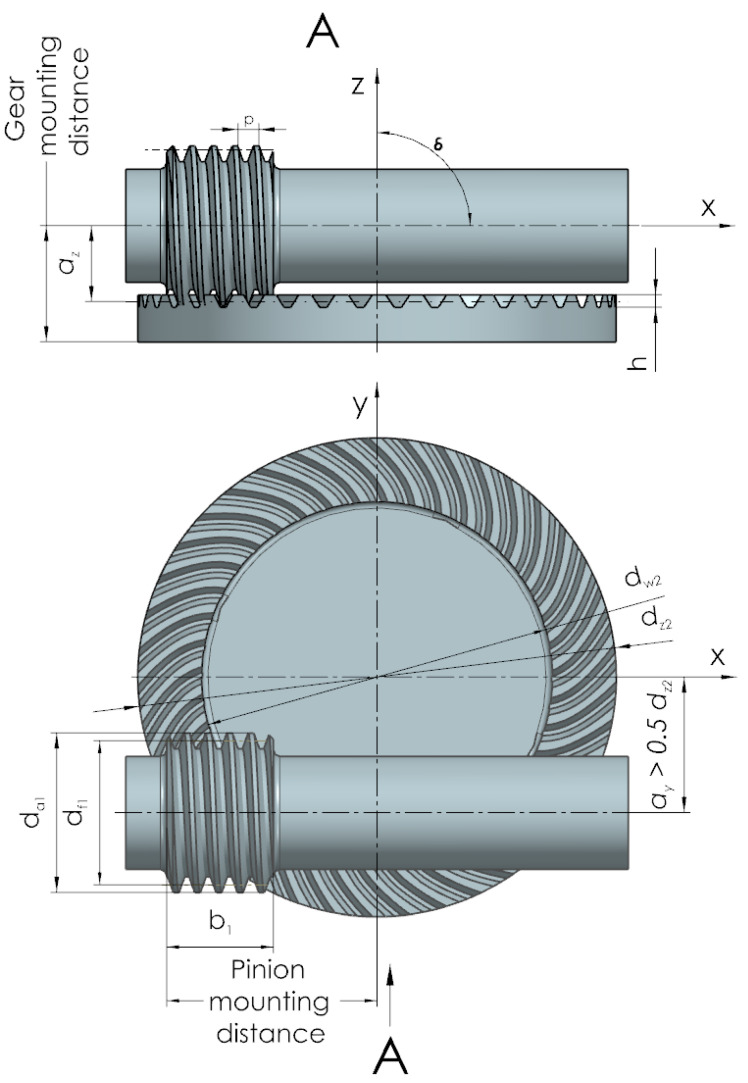
Basic dimensions of spiroid gear.

**Figure 4 sensors-22-08860-f004:**
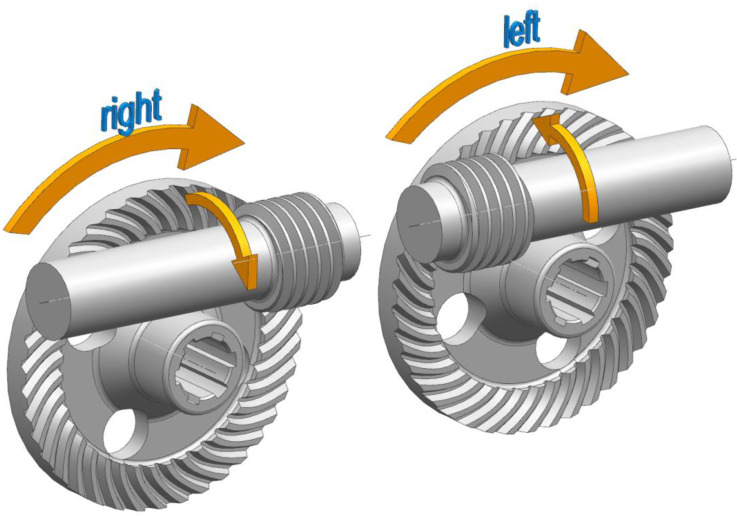
Types of spiroid gears depending on the direction of the worm shaft threads and the tooth line of the face gear.

**Figure 5 sensors-22-08860-f005:**
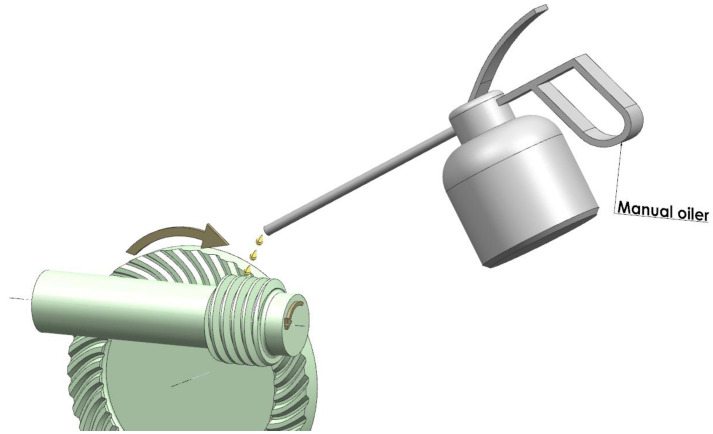
Intermittent lubrication of open gear, manual lubrication method.

**Figure 6 sensors-22-08860-f006:**
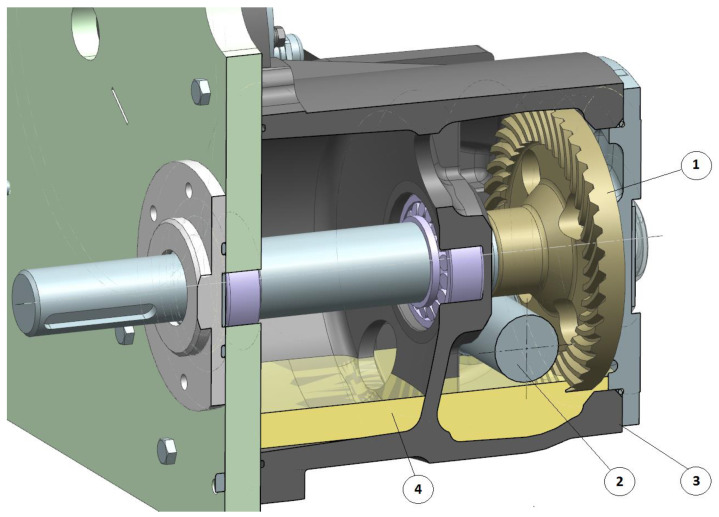
Immersion lubrication of spiroid gear; 1—face gear, 2—worm, 3—casing, and 4—oil.

**Figure 7 sensors-22-08860-f007:**
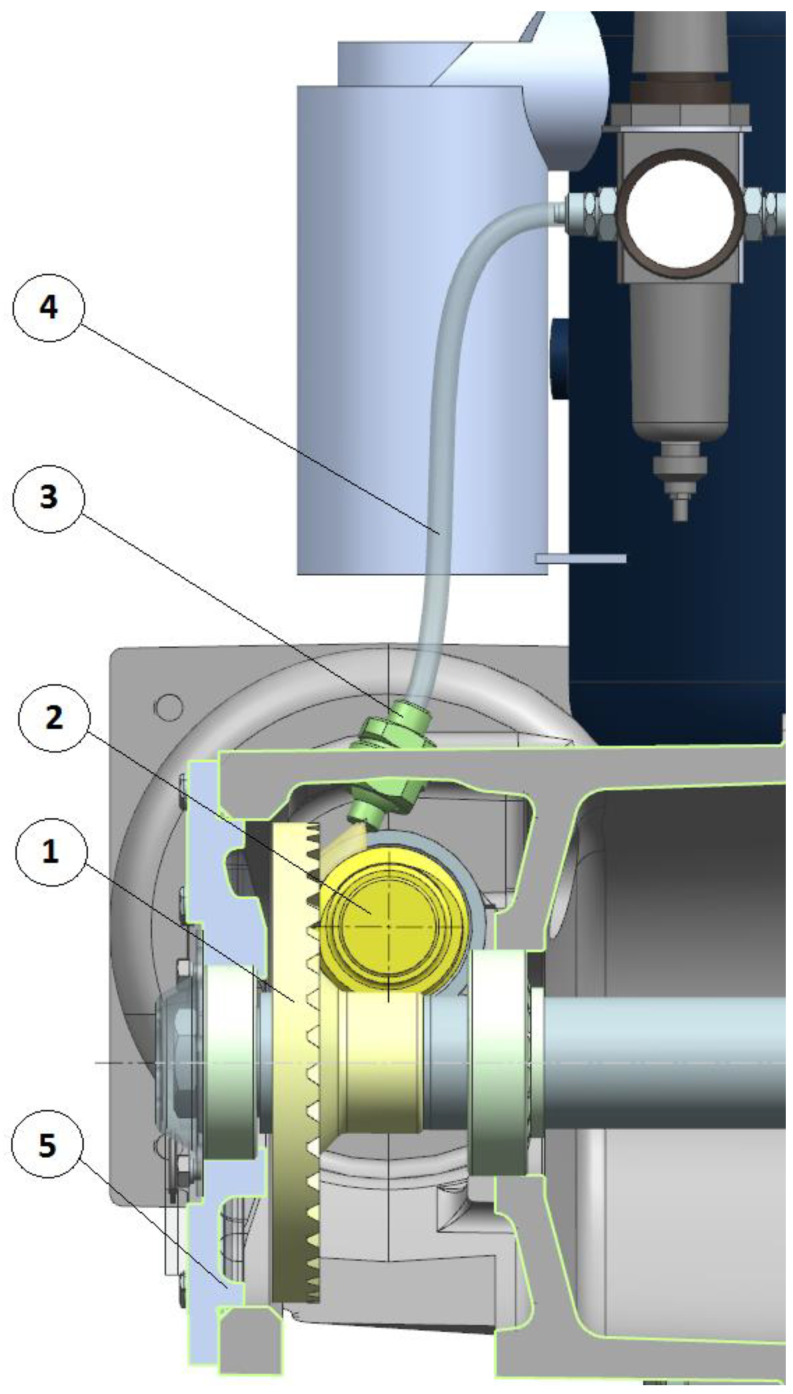
Spray (pressure) lubrication using spray nozzle. 1—face gear, 2—worm, 3—spray nozzle, 4—lubricant tube, and 5—casing.

**Figure 8 sensors-22-08860-f008:**
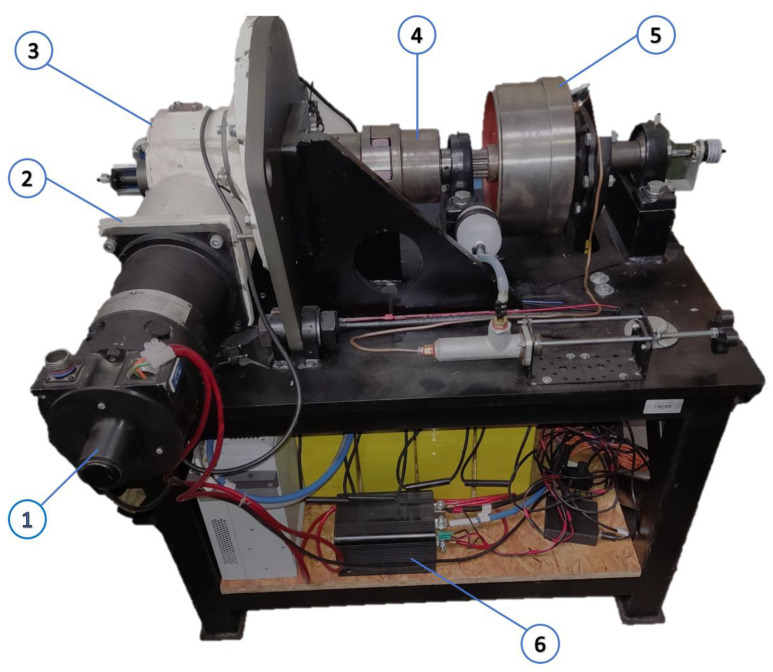
Test bench; 1—engine 1, 2—body, 3—spiroid gear, 4—coupling, 5—brake, and 6—measuring unit.

**Figure 9 sensors-22-08860-f009:**
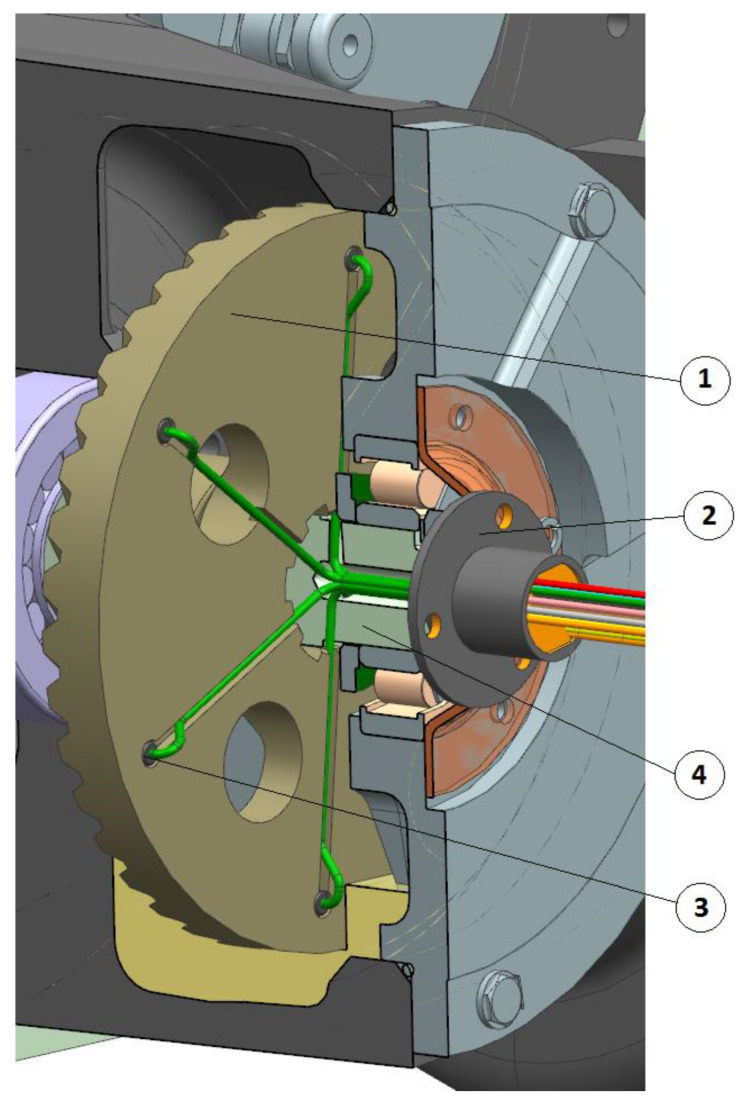
Location of thermocouples on the face gear; 1—face gear, 2—rotating joint, 3—temperature sensor, and 4—face gear shaft.

**Figure 10 sensors-22-08860-f010:**
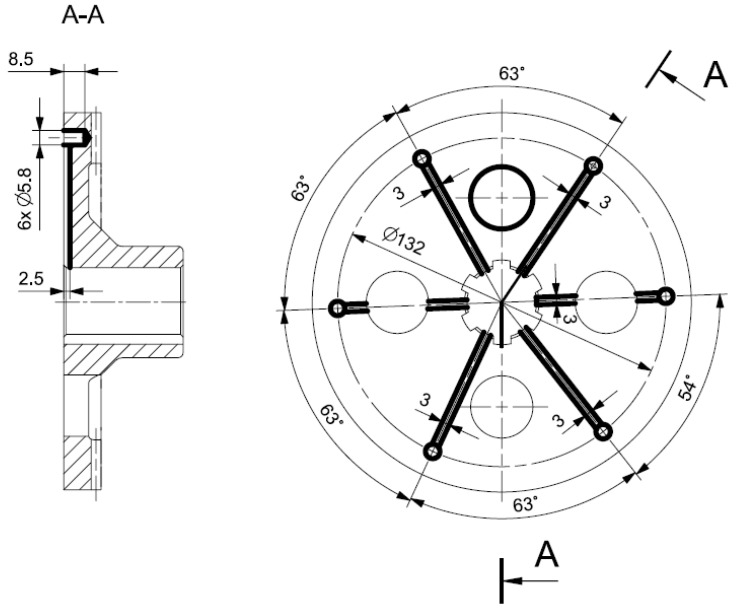
Location of sensors measuring the temperature of the face gear tooth in spiroid gear.

**Figure 11 sensors-22-08860-f011:**
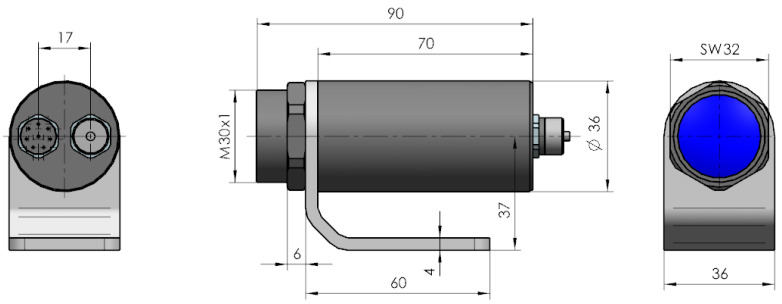
Dimensions of the Optris Xi 80 thermal imaging camera.

**Figure 12 sensors-22-08860-f012:**
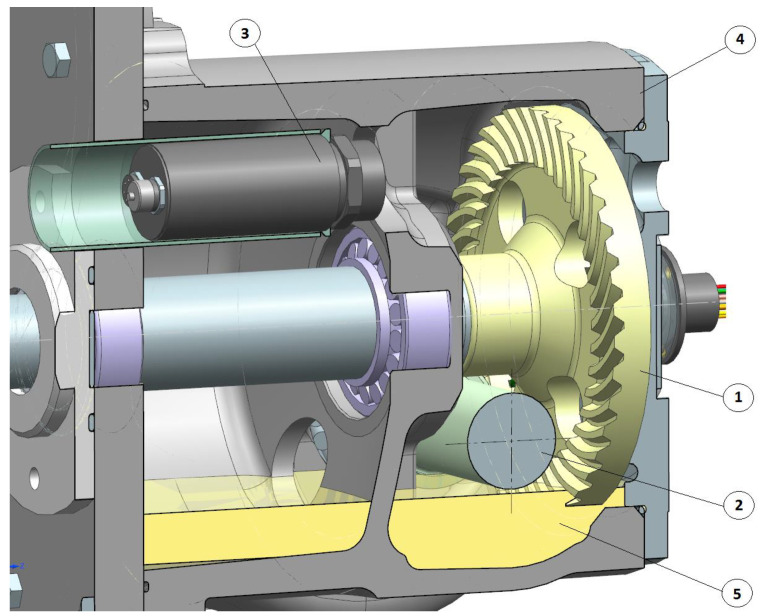
Location of the miniature thermal imaging camera Optris Xi80 in the body of closed gear; 1—face gear, 2—worm, 3—thermal imaging camera, 4—gear casing, and 5—oil.

**Figure 13 sensors-22-08860-f013:**
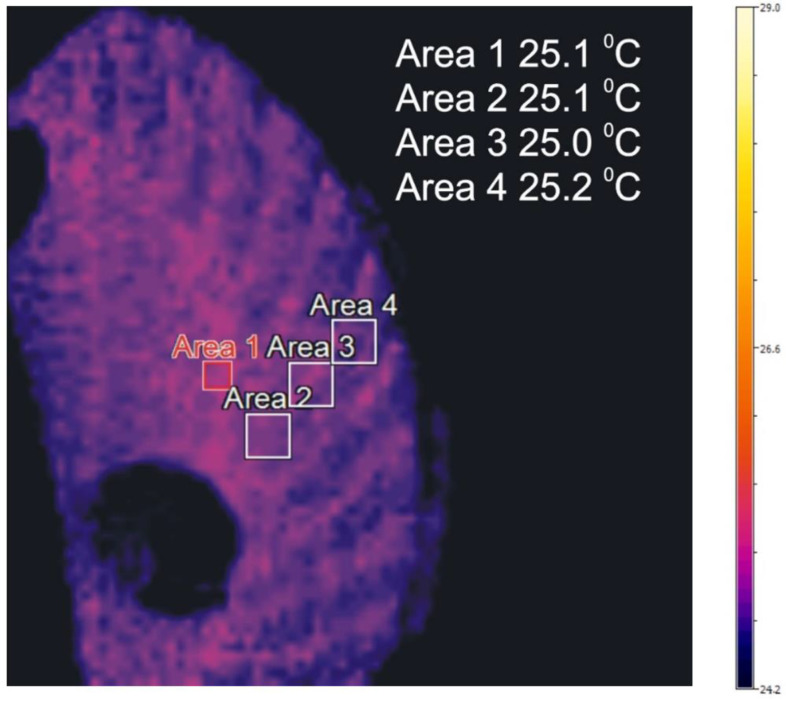
Distribution of temperature in the face gear, unloaded, in the reference temperature present the distribution of temperature for selected measurements from Table 3.

**Figure 14 sensors-22-08860-f014:**
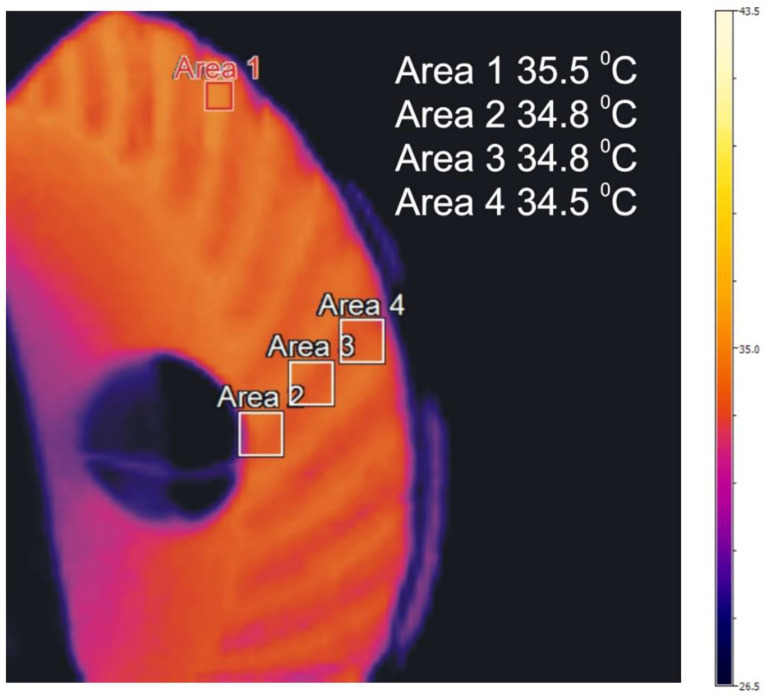
Distribution of temperature in the face gear, measurement number 1.

**Figure 15 sensors-22-08860-f015:**
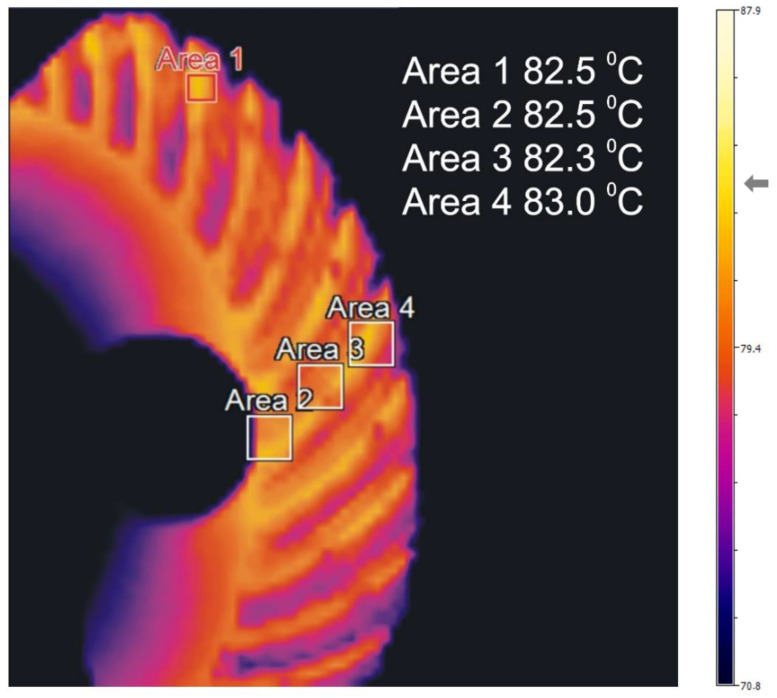
Distribution of temperature in the face gear, measurement number 8.

**Table 1 sensors-22-08860-t001:** Spiroid gear lubrication methods [28].

Peripheral Speed of the Worm (m/s)	Lubrication Method
up to 2	Manual lubrication
>2 to approx. 8	Immersion or spray lubrication
>8 to approx. 11	Spray (pressure) lubrication

**Table 2 sensors-22-08860-t002:** Basic operational parameters of the tested spiroid gear.

Test Bench Parameters		
Parameter	Designation	Value
Maximum engine torque	M_max_	41.6 Nm
Maximum engine speed	n_we_	3200 rpm
Gear ratio	*u*	20:1
Normal module	m_n_	3 mm
**Preset values**		
**Parameter**	**Designation**	**Value**
Torque on the input shaft	M_1_	40 Nm
Rotational speed on the input shaft	n_we_	500–1500 rpm
Oil type	GL-5	75 W90

**Table 3 sensors-22-08860-t003:** Measurement results.

Measured Values		Measurement Results							
Measurement Number		1	2	3	4	5	6	7	8
**n_1_ rpm—input speed**		500	1500	500	1500	500	1500	500	1500
**M_0_ Nm—resistance torque**		0	0	35	35	90	90	350	350
**Face gear rim**	**Area 1** **T_max_ °C**	35.5	41.6	44.2	50.8	56.6	60.8	64.3	82.5
	**Area 2** **T_max_ °C**	34.8	41.4	43.8	50.3	56.4	60.5	64.0	82.5
	**Area 3** **T_max_ °C**	34.8	41.4	44.0	50.4	56.6	60.5	64.1	82.3
	**Area 4** **T_max_ °C**	34.5	41.5	44.3	50.8	56.7	61.0	64.2	83.0
**T_ol_ °C—oil tank**		28.8	35.6	37.5	42.3	48.9	52.0	61.9	66.8

## Data Availability

Not applicable.

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
