# Peer review of "Measurement Method of Temperature of the Face Gear Rim of a Spiroid Gear"

_sensors, 2022, doi:10.3390/s22228860_

Round 1

Reviewer 1 Report (New Reviewer)

The paper is well, and the results are fascinating. I have the following observations.

The authors should state how accurate their study is?

Authors should give a comparison between their methods and already studied methods.

There are some minor typing mistakes which needs to be corrected.

Author Response

Thanks for your comments.

Please write down "Please see the attachment."

Reviewer 2 Report (New Reviewer)

This paper presents an innovative test bench for testing spiroid gears, the test bench measures the temperature distribution in the meshing zone of spiroid gears as well as the internal temperature of the gears during operation and the temperature of the lubricant in the oil tank. The manuscript is not ready for publication and requires a revision. The specific comments are listed as below:

1.The Introduction part, more related references of other gear temperature test methods need to be added.

2.In section 1 (page1, line 39), there are some errors in descripting the worm wheel outer diameter.

3.There are some errors in section 3(Lubrication of the mating toothed rims of a gear, line 138), such as Figure 5 is not consistent with the description of the text.

4.Table 3 is not clear, please check it.

5. It is suggested to develop a theoretical model to validate the experimental results.

Author Response

Thanks for your comments.

Please write down "Please see the attachment."

Reviewer 3 Report (New Reviewer)

Comments to the Author

1.     Fig.2 is not required.

2.     Journal Reference numbers are not in order in the entire paper.

3.     Grammar and English must be improved. Example Check line 76-77.

4.     Some References lack information such as volume numbers and page numbers.

5.     What is EP in line 127. Also, it must be appropriate to use elastohydrodynamic lubrication instead of friction.

6.     Include the specific reasons for the selection of test bench parameters and preset values. Why was the test carried out for 4 torques and 2 speeds? Also, mention the standard followed in the test if there is any.

7.     Table 3 shows only the temperature measured from the thermal image camera. What about the temperature measured from the thermocouple?

8.     Include the figures for the distribution of temperature in the face gear for all the other test numbers 2,3,4,5,6.  

9.     Discussion on the measurement values can be elaborated.   Line 269 – 275. 

Author Response

Thanks for your comments.

Please write down "Please see the attachment."

This manuscript is a resubmission of an earlier submission. The following is a list of the peer review reports and author responses from that submission.

Round 1

Reviewer 1 Report

           This study seems to be good and address the current need in the mobility segment. However, with the way that currently manuscript is structured, it is hard to see if this work has any novelty or if there is any specific finding from this study that differentiates it from previous work. This paper still has room for improvement in the introduction and discussion sections. I would recommend that authors take benefit of recent works that has been done in this field and strengthen the literature review and discussion section. My sense of observation is the paper can be accepted after the modifications listed below:

1.     There is no novelty in the result output. It can be noticed that the abstract just focusing on methodology. Please include some results in the abstract.

2.     The literature review is poorly written. You just listed the related studies. However, why you choose to do this study is not mentioned. For instance they can refer and include the following paper for the guidance in writing the introduction part.

3.   Please read the paper carefully for English language style and accuracy, and make appropriate corrections and changes. With assist of professional experts rectify the grammatical and vocabulary error in the paper wherever necessary.

4.     It is recommended that each referenced work is accompanied by a brief description of the key results and main conclusions. In this way the inclusion of the cited work in the manuscript can be better justified.

5.     The results section has to be modified and in the revised manuscript more results can be included. At the end of the paper a comparison with other similar type of works will improve the quality of the paper.

6.     In the conclusions, in addition to summarizing the actions taken and results, please do explain their significance, also add the quantifiable results for all the parameters you have studied. Please give main contribution of this study the end of conclusion section for further studies.

Reviewer 2 Report

In the paper, the authors built a spiroid gear test bench to measure the temperature field. The conclusion is noticeable, and the organization of the article is very poor. The paper is not recommended for publication.

(1) The description of the abstract is nothing new. It does not highlight the innovations and main conclusions of the work.

(2) The number of keywords is no enough.

(3) Figures 1 and 2 are not required and are recommended to be deleted.

(4) The writing is poor.

(5) The citation of references is confused. In addition, the description of the existing literatures is inadequate.

(6) In line 132, the symbol ‘m’ is inconsistent with the description in Table 2.

(7) The content of Section 4 is inconsistent with the topic of the paper. Meanwhile, chapter number ‘4’is repeated.

(8) The results are simply described without detailed discussion. In addition, the research content of the paper is insufficient.

Reviewer 3 Report

This manuscript has great innovative significance in investigating method of measurement the temperature of the face gear rim of a spiroid gear. The work can arouse wide interests of researchers in design and preparation of new functional parts. The manuscript is interesting. In my frank opinion, the manuscript should be deserved for its final publication in high-level Journal-sensors. The main reasons are as follows:

1. What does temperatureof mean in the title? Please note the format.

2. At first, the English ABSTRACT should be revised, and a unified simple present tense should be used.

3. The research significance and future work should be described in the final stage of the abstract.

4. Aims need to be concisely stated and added at the end of introduction. Not only what was done/investigated, but why.

5. Under normal conditions, in conclusion section, important conclusions should be elaborated point by point for brevity and prominence, such as a) … … b) … … c) … ….

6. And also in the last point future research work should be given in conclusion section.

Reviewer 4 Report

The presented scientific contribution entitled Method of measurement of the temperature of the face gear rim of a spiroid gear deals with a very current topic, but overall, the described issue is very insufficiently processed. Based on the following comments, I decided to reject the submitted contribution:

·      -  A very superficially processed introduction, without using references published in recognized journals. There are only three references in the introduction.

·    -     Not all references are listed in the text of the scientific contribution. Specifically, references 6, 8 and 29 are missing.

·      -   Chapter number 4 is mentioned in the contribution twice. I assume that the first mentioned does not belong to this contribution.

·       -  Insufficiently explained and described methodology of the experiment.

·       - Why weren't the measured temperature values from the six test points located on the face gear rim evaluated?

·      -   Verification measurements are insufficient, they are not repeated, they are not statistically evaluated.

·     -    The conclusion only partially corresponds to the content of the scientific contribution.

·      -   I have doubts about the cited references, because most sources are cited in three lines.